

# Applicability of VGGish embedding in bee colony monitoring: comparison with MFCC in colony sound classification

Nayan Di[1,2], Muhammad Zahid Sharif[1,2], Zongwen Hu[3,4], Renjie Xue[1,2] and Baizhong Yu[1,2]

[1] Anhui Institute of Optics and Fine Mechanics, Hefei Institute of Physical Science, Chinese Academy of Sciences, Hefei, China
[2] University of Science and Technology of China, Hefei, China
[3] Eastern Bee Research Institute, College of Animal Science and Technology, Yunnan Agricultural University, Kunming, China
[4] The Sericultural and Apicultural Research Institute, Yunnan Academy of Agricultural Sciences, Mengzi, China

## ABSTRACT

**Background**. Bee colony sound is a continuous, low-frequency buzzing sound that varies with the environment or the colony's behavior and is considered meaningful. Bees use sounds to communicate within the hive, and bee colony sounds investigation can reveal helpful information about the circumstances in the colony. Therefore, one crucial step in analyzing bee colony sounds is to extract appropriate acoustic feature. **Methods**. This article uses VGGish (a visual geometry group—like audio classification model) embedding and Mel-frequency Cepstral Coefficient (MFCC) generated from three bee colony sound datasets, to train four machine learning algorithms to determine which acoustic feature performs better in bee colony sound recognition. **Results**. The results showed that VGGish embedding performs better than or on par with MFCC in all three datasets.

Corresponding author
Nayan Di, dny_yan@126.com

## INTRODUCTION

Honey bees play an essential role in agricultural production, responsible for pollination of almost 90% of the world's commercial pollination services (*Klein et al., 2007*) and pollination of natural habitats (*Hung et al., 2018*). As a vital node of the agriculture section, it is essential to ensure that the bee colonies can provide service. To save human resources and reduce disturbance to bee colonies, a non-invasive or minimally invasive method that can detect the intra-colonial condition of the hive without disturbing the colony is a consensus among researchers and practitioners (*Bencsik et al., 2011*; *Qandour et al., 2014*; *Gil Lebrero et al., 2016*). Since the colony weight, temperature, humidity, gas concentration and sound in the hive are relatively stable, much information about the status of the colony can be learned by monitoring the indicators and establishing the association between these indicators (*Meikle & Holst, 2015*; *Ferrari et al., 2008*; *Murphy et al., 2015*; *Braga et al., 2020*). Among these indicators, beehive sound is critical. Bee buzzing carries information

on colony behavior and phenology. Honey bees emit specific sounds when exposed to stressors such as pest infection (*Qandour et al., 2014*), airborne toxicant (*Zhao et al., 2021*), swarming detection (*Ferrari et al., 2008*; *Zlatkova, Kokolanski & Tashkovski, 2020*), and failing queens (*Cejrowski et al., 2018*; *Soares et al., 2022*). Using both statistical and Artificial Intelligence (A.I.) analysis of colony sounds, *Bromenshenk et al. (2009)*, in their patents (*Bromenshenk et al., 2009*) and in their review article (*Bromenshenk et al., 2015*) showed that their A.I. could detect a diverse variety of chemicals and eight colony health variables, by simply putting a microphone into the bottom of a beehive and recording bee colony sounds for 30 or 60 s. In 2019, they released a cellphone app (Bee Health Guru) that can run the diagnostic programs, record and analyze the results, and upload the data, visual inspections, and app analyses to a cloud-based site, which automatically generates a report with the GPS location shown on a map. Currently, the app is being calibrated for a variety of phone operating systems for bee sounds from around the world (http://www.beehealth.guru).

One of the critical phases in analyzing the bee colony sound would be extracting appropriate feature for machine learning or deep learning algorithms. Traditionally we use frequency domain or time domain feature of sound, such as soundscape indices and low-frequency signal features (*Sharif et al., 2020*). MFCC is one of the most commonly used features in bee colony sound analyzing (*Kim, Oh & Heo, 2021*; *Soares et al., 2022*). It is characterized by using a set of critical coefficients to create Mel cepstrum, which makes its cepstrum more similar to the nonlinear human auditory system (*Muda, Begam & Elamvazuthi, 2010*). Due to the nonlinear correspondence between Mel frequency and Hz frequency, the calculation accuracy of MFCC decreases with the increase of frequency. This characteristic makes MFCC more suitable for bee colony sound than other feature extraction methods in the past because the sound signal in the colony is concentrated in the low-frequency part (*Dietlein, 1985*).

Thanks to the rapid development of artificial intelligence, convolutional neural net (CNN) and recurrent neural networks (RNN) have been applied in audio recognition (*Kumar & Raj, 2017*). Experimental results showed that the recognition method based on CNN is prior to or on par with the method based on machine learning models in beehive audio classifying (*Kulyukin, Mukherjee & Amlathe, 2018*). Visual Geometry Group (VGG) is one of the most popular CNN models. Simonyan and Zisserman proposed it in 2014 and is named after the Visual Geometry Group (*Simonyan & Zisserman, 2014*). VGGish is a TensorFlow definition of a VGG-like audio classification model. The VGGish model is a derivative network of the VGG network trained on a large YouTube dataset (*Gemmeke et al., 2017*). Its structure is consistent with VGG11, including eight convolutional layers, five pooling layers, and three fully connected layers. Each convolutional layer uses a 3x3 convolution kernel. VGGish converts audio input feature into a semantically meaningful, high-level 128-dimensional embedding, which can be fed as input to a downstream classification model. On account of the scale and diversity of the YouTube dataset, the resulting acoustic feature are both very general and of high resolution, placing each audio sample in a high-dimensional feature space that is unlikely to show ecosystem-specific bias. This 128-dimensional embedding characteristic is helpful in various identification

contexts, including monitoring anomalous events in an ecosystem (*Sethi et al., 2020*) and sound-based disease detection (*Shi et al., 2019*).

In this article, we contribute to the body of research on audio beehive monitoring by comparing VGGish embedding and standard MFCC in classifying audio samples from microphones deployed inside beehives. We tested the VGGish embedding and MFCC on three different classification tasks and compared these two features using four machine-learning algorithms.

In particular, section two will describe the hardware and software configuration to obtain bee colony sound and report the detail of the three bee colony datasets we used in this article. Section three will give the performance of VGGish embedding and MFCC in bee colony sound classification, as well as the effects of different dimensional reduction algorithms. Section four will report conclusions and a future perspective.

## MATERIALS AND METHODS

### Hardware

The hardware and software systems for obtaining bee colony sound are as follows: a microphone inside the beehive (PCK200, Takstar, Guongdong, China) was placed about 15 cm from the bottom. The microphone has a frequency range of 30 Hz to 20 kHz and a sensitivity of $-35$ dB. A digital sound card (UM2, Behringer, Willich, Germany) was used to convert the analogue signal into a digital signal. The digital signal was transmitted to a personal computer (HP 2170p, Windows 7), The software Audacity was used to record the sound, and the sound sampling rate was set to 44.1 kHz, mono. Sound files were saved on the hard disk in .wav format. The hardware structure is illustrated in Fig. 1.

### Audio data

The experiment was carried out at the Sericulture and Apiculture Research Institute of Yunnan Academy of Agricultural Sciences (23.5144N, 103.4043E) from November 2020 to June 2021. The institute is located in Caoba Town, Mengzi City, Yunnan Province, China. We collected three collections of honeybee (*Apis cerena*) colony sounds and named them dataset one, two and three, respectively. A detailed description of these datasets is given below. Every bee colony resided in a typical wood beehive with a 10-month-old queen. All the bee colonies were healthy without any sign of attack by pests, emerging diseases, and viruses.

#### Dataset one

Dataset one contains the colony sound of three experimental groups. Each group was treated with unique odorous compounds.

Honeybees were trained with syrup to visit artificial feeding sites approximately 200 m from the hive. A feeder containing 50% sucrose solution was placed 5 m from the hive, and the marked foragers were caught in a glass tube at the hive entrance. The foragers were gently let out to the feeder. When the foragers had eaten enough syrup, they returned to the hive after hovering over the feeder a few times. This was repeated several times, and when visited by many foragers, the feeder was slowly placed approximately 10 m from the hive,

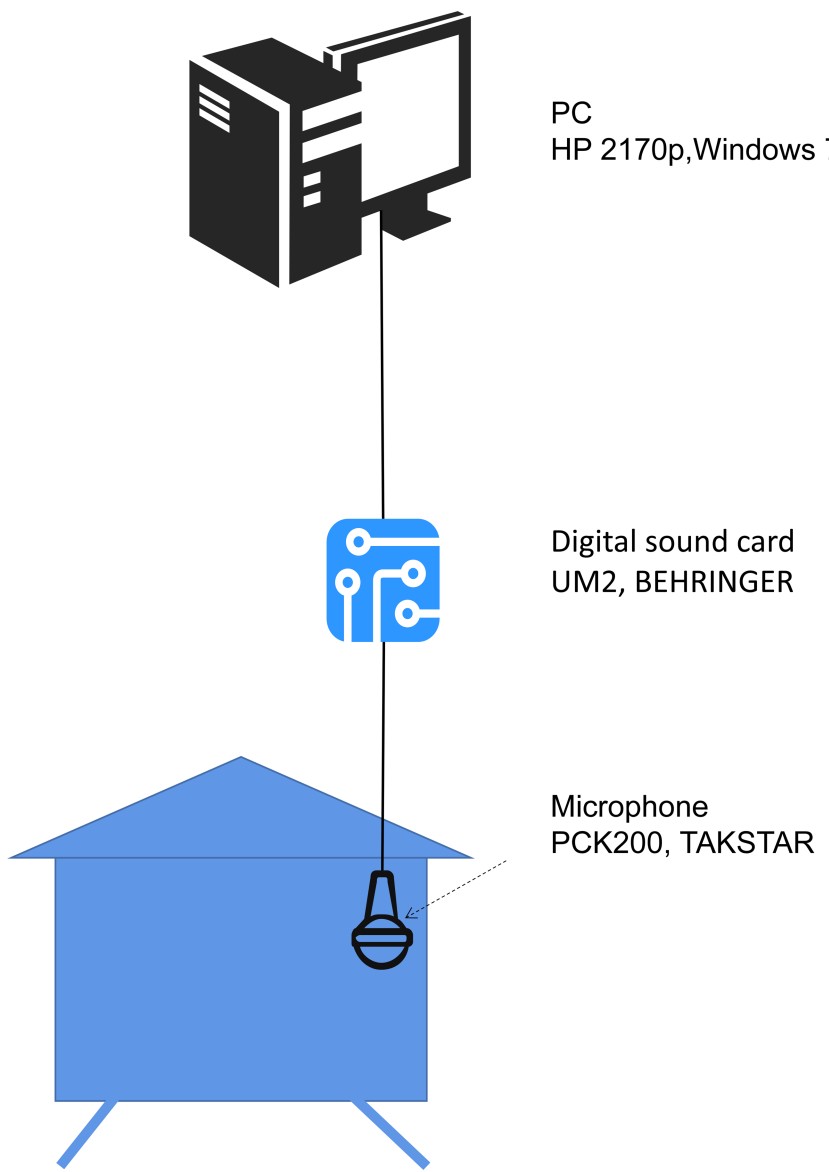

PC
HP 2170p,Windows 7

Digital sound card
UM2, BEHRINGER

Microphone
PCK200, TAKSTAR

**Figure 1** **The hardware system used to obtain bee colony sound.** The microphone is placed inside the beehive. The sound signal captured by the microphone is converted to a digital signal by the digital sound card, then transmitted to the PC and saved on a hard disk for further analysis.

and so on, gradually moving the feeder to 200 m. When a large number of marked bees were visiting the feeder, the sound inside the colony was recorded for 10 min. In addition to pure syrup, we fed the bees syrup containing ethyl acetate and syrup containing acetone, respectively. Before changing the compounds added to the syrup, we stopped feeding for two days, waiting for the colony to be depleted of food and odors before starting another treatment. The sound files were collected from three different colonies, each colony with two frames, the number of recordings and duration were shown in Table 1. The colony sound files were collected during winter from November 2020 to January 2021, and very
**Table 1 An overview of the datasets collected in order to identify compounds in nectar and queen's presence.** "Scenario "N recordings" denotes the number of individuals with buzzing sounds recorded; "Total duration" represents the total recording time in each case; N colonies denotes the number of colonies in which we recorded sounds; N frames represent the colony size.

| Datasets | Scenario | N colonies | N frames | N Recordings | Total duration |
|---|---|---|---|---|---|
| Dataset one Identify compounds | Blank | 3 | 2 | 6 | 50 min |
|  | Acetone | 3 | 2 | 9 | 90 min |
|  | Ethyl acetate | 3 | 2 | 11 | 111 min |
| Dataset two Identify queen state | Blank | 2 | 6 | 12 | 131 min |
|  | New queen pupa | 2 | 6 | 9 | 101 min |
|  | New queen | 2 | 6 | 3 | 23 min |
| Dataset three Identify colony size | C2 | 2 | 2 | 2 | 12 min |
|  | C4 | 2 | 4 | 2 | 15 min |
|  | C6 | 2 | 6 | 2 | 29 min |

few food sources were available outside. In this way, the artificial food source we provide may be the only food sources for honeybees.

This dataset contains the colony sound of three experimental groups, which were treated with unique odorous compounds at a mass ratio of 0.1% in 50% (w/w) sucrose solution, sucrose solution with 50% concentration was used as blank control. The colony sound was labeled 'blank,' 'acetone,' and 'ethyl,' respectively.

### Dataset two

Dataset two collects bee colony sounds concerning the queen's status. The object is to use the colony sound to detect whether there is a queen pupa and whether the pupa has hatched. This dataset includes honey bee sounds under three scenarios.

This work was carried out in June 2021, alternating between spring and summer. It simulated the occurrence of a new queen cell in the colony before swarming. We selected two groups of healthy and strong colonies of *Apis cerana*, each with six frames of honeybees and a normal breeding queen. In the first scenario, we caged the queen and collected colony sounds. In the second stage, we introduced a mature queen pupa into this colony. The original queen was still in the cage and, therefore, would not attack the new queen pupa. Collecting sound data began after a day. In the third stage, we opened the hive every night, checked the pupa condition, and recorded the next day after the new queen emerged. All recordings started around 11:00 am. In this way, we obtained colony sounds in three different queen states. They were labeled as 'blank,' 'queen pupa,' or 'new queen.'

### Dataset three

This dataset contains sounds from bee colonies of different colony sizes. We investigated six bee colonies, including two colonies with two frames, two with four, and two with six. The bee colony sound was recorded at 9:00 am for about three to ten minutes in each of the colonies, and the recorded sound files were labeled as 'C2', 'C4', and 'C6,' respectively. We reckon the number of bees by weighing the colony. The weight of an empty hive is measured first, then the whole swarm of bees is shaken off into the empty hive, and the
**Table 2 The size of each colony used in dataset three.** "N frames" denotes the number of frames in the colony; "Total bee weight(Kg)" represents the total weight of each colony; "N worker bees denotes the approximate number of worker bees in each colony.

| Colony | N frames | Total bee weight (Kg) | N worker bees |
|--------|----------|-----------------------|---------------|
| 1# | 2 | 0.723 | 8,670 |
| 2# | 2 | 0.685 | 8,213 |
| 3# | 4 | 1.010 | 12,110 |
| 4# | 4 | 1.095 | 13,129 |
| 5# | 6 | 1.650 | 19,784 |
| 6# | 6 | 1.580 | 18,945 |

mass is measured again. The mass difference obtained is the total weight of the swarm. We estimated the number of bees per colony based on the average honeybee weight, which was $83.4 \pm 10.2$ mg (Table 2).

## Data processing

The data processing was based on python 3.5.1 and Scikit-learn 1.0.2 (*Pedregosa et al., 2011*).

### Feature extraction

*VGGish Embedding.* The audio sample was first split into segments of 0.96s. Each 0.96s segment was first resampled to 16 kHz using a Kaiser window, and a log-scaled Mel-frequency spectrogram was generated (96 temporal frames, 64 frequency bands). Each audio sample was then passed through CNN from Google's AudioSet project (*Gemmeke et al., 2017*; *Hershey et al., 2017*) to generate a 128-dimensional embedding of the audio. Figure 2 shows the structure of the VGGish network and the workflow of extracting VGGish embedding.

*Mel-frequency Cepstral Coefficient (MFCC).* MFCCs are based on the known variation of the human ear's critical bandwidths with frequency. The MFCC technique uses two types of filters: linearly spaced and log arithmetically spaced. The signal is expressed in the Mel frequency scale to capture the phonetically important characteristics of speech. This scale has a linear frequency spacing below 1,000 Hz and a logarithmic spacing above 1,000 Hz. The MFCC extraction procedures are as follows: windowing the sound signal, applying the FFT (Fast Fourier Transform), taking the log, and then warping the frequencies on a Mel scale, followed by applying the inverse DCT (Discrete Cosine Transform). The 13-dimensional MFCCs will be combined with the first-order difference coefficients and second-order coefficients difference to get the 39-dimensional MFCCs (*Davis & Mermelstein, 1980*).

A block diagram of the structure of an MFCC processor is given in Fig. 3.

### Dimension reduction

Since the features extracted from the raw data are high-dimensional, it is not conducive to visualization. It is necessary to use the technique for dimensionality reduction to get 2D points from a high-dimensional input vector.

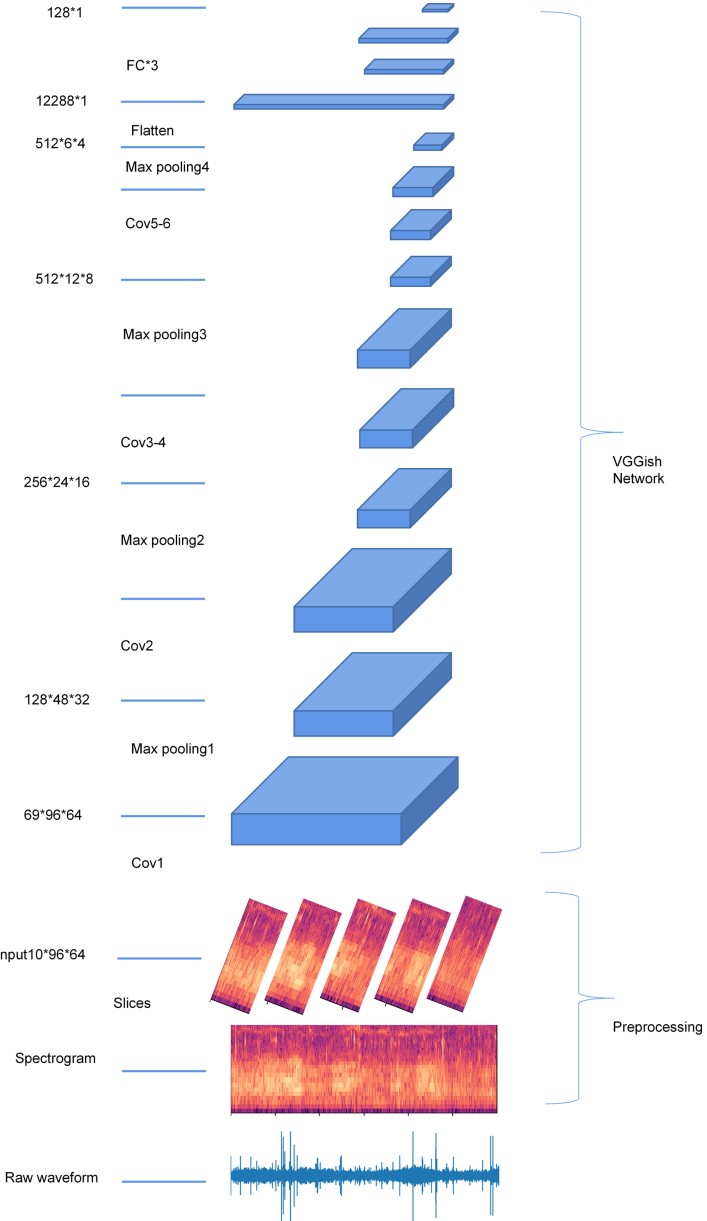

**Figure 2** An overview of the structure of the VGGish network.

To estimate the impact of dimension reduction, we experimented with the following dimensionality reduction algorithms: (R1) uniform manifold approximation and projection (UMAP). UMAP works by learning approximate manifolds from higher dimensional Spaces and mapping them into lower dimensional Spaces (*McInnes, Healy & Melville, 2018*); (R2) t-distributed stochastic neighbor embedding (t-SNE) (*Vander Maaten & Hinton, 2008*). This technique is a variation of Stochastic Neighbor Embedding (*Becht et al., 2019*; *Diaz-Papkovich et al., 2019*).

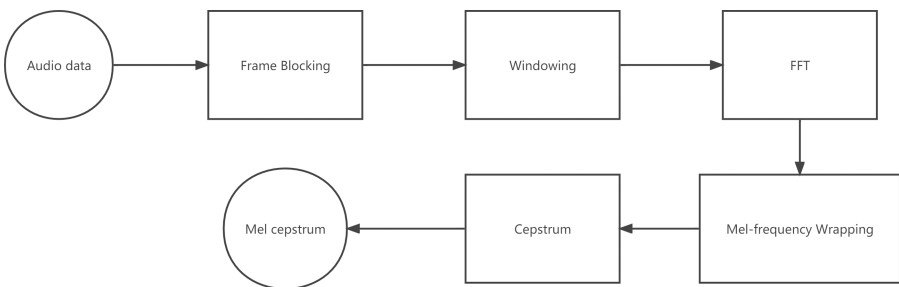

**Figure 3** **Block diagram of the MFCC processor.**

The multidimensional colony sound features were narrowed down to two by the two algorithms. Machine learning algorithms then classify the reduced feature set.

### Training classifiers

In this article, we trained four well-known machine learning (ML) algorithms, namely decision tree (DT), K-nearest neighbors (KNN), support vector machine (SVM) classification, and random forests (RF). DT is a tree-structured classifier. The internal nodes represent the features. The branches represent the rules, and each leaf node represents the outcome. KNN (*Altman, 1992*) is a supervised learning model. A majority vote classifies its neighbors in vector space, and the data is assigned to the class with the nearest neighbors. SVM classification (*Hong & Cho, 2008*) aims to create the best decision boundary (which is called a hyperplane) that can segregate n-dimensional space into classes so that the new data point can be put into the correct category. RF is a classifier that contains a bunch of decision trees (*Breiman, 2001*). It takes the prediction from each tree and predicts the final output based on the majority votes of predictions from those decision trees.

We trained all four models on the same feature vectors automatically extracted from the raw audio files in three bee colony datasets. The following feature: (F1) VGGish embedding; (F2) Mel frequency cepstral coefficients (MFCCs) (*Davis & Mermelstein, 1980*) are used in training all four models. We used the mean of the test accuracy as a summary of the model's performance. Then the paired Student's $t$-test was used to check if the difference in the mean accuracy between the two models is statistically significant.

The labeled features were split into a training set (70%) and a testing test (30%) with the training_test_split procedure from the Python sklearn.model_selection library (*Pedregosa et al., 2011*). All these classification models were trained with the training set on an Intel Xeon E5−2676 V3@2.40 GHz x 12 processor with 64 GiB of RAM and 64-bit Windows 10.

### Model evaluation

Classification accuracy and F1 score were used to evaluate the performance of the ML models. The classification accuracy is the percentage of correct predictions. The F1 score integrates information regarding both precision and recall (*Chinchor & Sundheim, 1993*). The balanced accuracy of the classifier on the test set was reported as an average F1 score for each class to account for sample-size imbalances among classes.

The data processing work-flow is presented in Fig. 4.

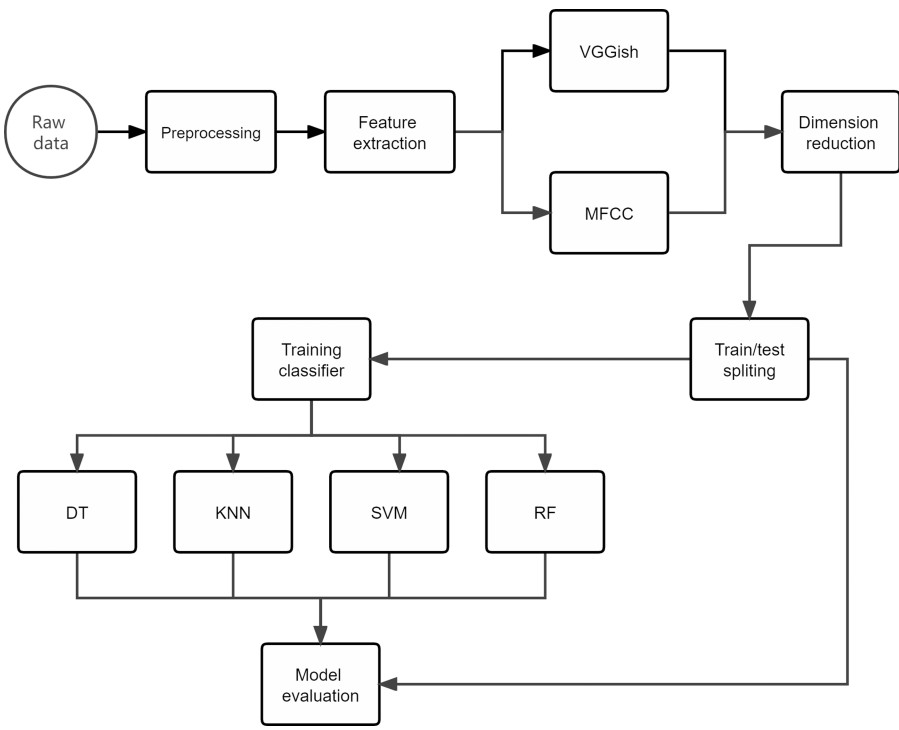

**Figure 4** **Overview of the approach adopted for the acoustic classification of beehive sounds workflow.** The original audio files (.wav format) containing recordings of beehive sounds were manually classed into corresponding scenarios. Then, the MFCC and VGGish embedding were used to extract the audio features, respectively. Dimensionality reduction was performed using the UMAP method for the two sets of feature data. After that, the resulting data set was split into 70% for the training/development set and 30% for the testing data set. Finally, the test data set was used to evaluate the performance of the classifiers in correctly assigning the beehive sound to the respective scenario.

# RESULTS

## The performance of models on dataset one

### *Audio signal*
Two different compounds were added separately into the sucrose solution. Figure 5 presents the log spectrogram of the bee colony sound. We can see that: (1) after being treated with a compounds-sucrose solution, the low-frequency sound in the bee colony increased; (2) the bee colony sound increased more significantly when feeding with the acetone-sucrose solution than when feeding with ethyl-sucrose solution, and there was a significant increase in bee colony sound around 130 Hz.

### *Dimensional reduction of audio feature*
Figure 6 shows the output of VGGish embedding and MFCC dimensionality reduction in dataset one. In the two-dimensional diagram, it is evident that the MFCC overlaps after dimensionality reduction, while the VGGish embedding can better distinguish the sound in these three situations.

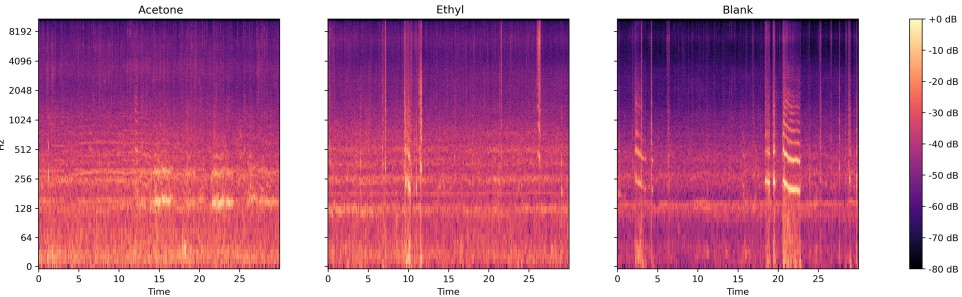

**Figure 5  Log spectrum of bee colony sounds from dataset one.** Left: Acetone (treated with acetone-sucrose solution); Middle: Ethyl (treated with ethyl acetate-sucrose solution); Right: Blank (treated with sucrose solution).

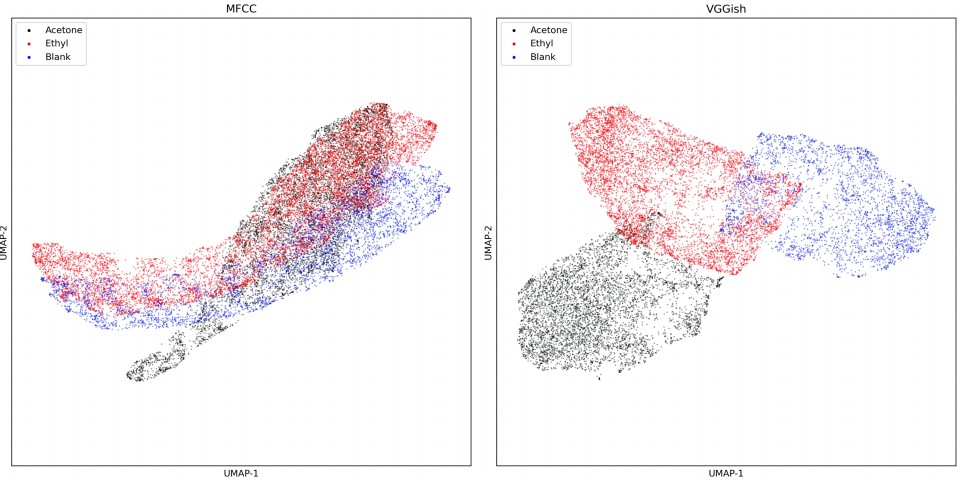

**Figure 6  UMAP dimension reduction of sound features from dataset one.**

**Table 3  Accuracy of machine learning models using different colony sound features on three colony sound datasets.**

| Datasets | Dataset 1 | | | | Dataset 2 | | | | Dataset 3 | | | |
|---|---|---|---|---|---|---|---|---|---|---|---|---|
| Algorithm | KNN | DT | RF | SVM | KNN | DT | RF | SVM | KNN | DT | RF | SVM |
| VGGish | 94.79% | 93.45% | 94.43% | 91.56% | 86.58% | 85.14% | 85.94% | 81.46% | 91.08% | 88.81% | 89.23% | 89.15% |
| MFCC | 69.09% | 66.28% | 69.17% | 68.29% | 90.48% | 88.45% | 89.95% | 87.25% | 66.04% | 65.78% | 65.13% | 68.05% |

### Model evaluation

Tables 3 and 4 summarize the results of four machine learning methods. VGGish embedding performs significantly better than the MFCC ($P < 0.005$) and shows an advantage of about 30% over MFCC in all four machine learning methods, among which KNN performs best, achieving an accuracy of 94.79%.

**Table 4  F1-score of machine learning models using different colony sound features on three colony sound datasets.**

| Datasets | Dataset 1 | | | | Dataset 2 | | | | Dataset 3 | | | |
|---|---|---|---|---|---|---|---|---|---|---|---|---|
| Algorithm | KNN | DT | RF | SVM | KNN | DT | RF | SVM | KNN | DT | RF | SVM |
| VGGish | 94.79% | 93.45% | 94.42% | 91.55% | 86.58% | 85.17% | 85.93% | 81.49% | 91.06% | 88.82% | 89.21% | 89.03% |
| MFCC | 68.24% | 66.32% | 68.49% | 65.26% | 90.13% | 88.44% | 89.63% | 85.41% | 65.73% | 65.74% | 64.80% | 65.09% |

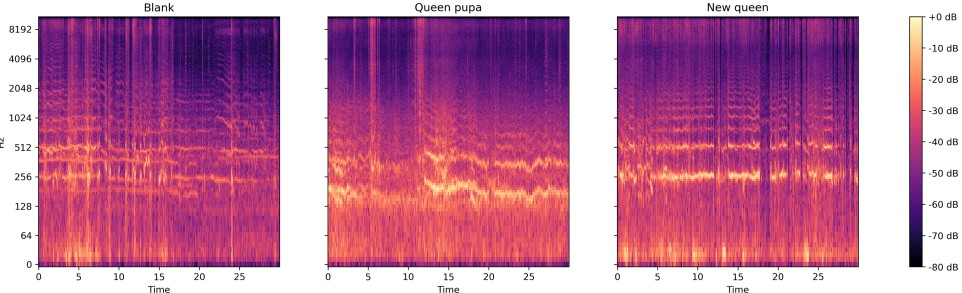

**Figure 7  Log spectrum of bee colony sounds of dataset two.** Left: normal situation; Middle: Queen pupa inside colony; Right: new queen emerged (two queens in the colony).

## The performance of models on dataset two

### *Audio signal*

From the log spectrogram of the bee colony sound (Fig. 7), the colony with a queen pupae seemed more active than the colonies in the other two conditions. The signal around 250 Hz and 500 Hz are stronger in the sound collection 'Queen pupa' and 'New queen' than in the sound collection 'Blank.'

### *Dimensional reduction of audio feature*

Compared with the MFCC dimensionality reduction diagram (Fig. 8), the scatter plot of VGGish embedding after dimensionality reduction has less overlap.

### *Model evaluation*

The MFCC performs slightly better than VGGish embedding and shows an advantage of about 4 percent of four machine learning methods (Tables 3 and 4). Still, the difference was not statistically significant ($P > 0.05$). Moreover, KNN performed best, and achieved an accuracy of 90%.

## The performance of models on dataset three (identifying colony size)

### *Audio signal in dataset three*

This dataset includes bee colony sounds from three colony sizes: C2) bee colony size of about 8,500 worker bees; C4) bee colony size of about 12,000–13,000 worker bees; C6) bee colony size of about 19,000–20,000 worker bees. Figure 9 presents the log spectrogram of the bee colony sound signals in this dataset.
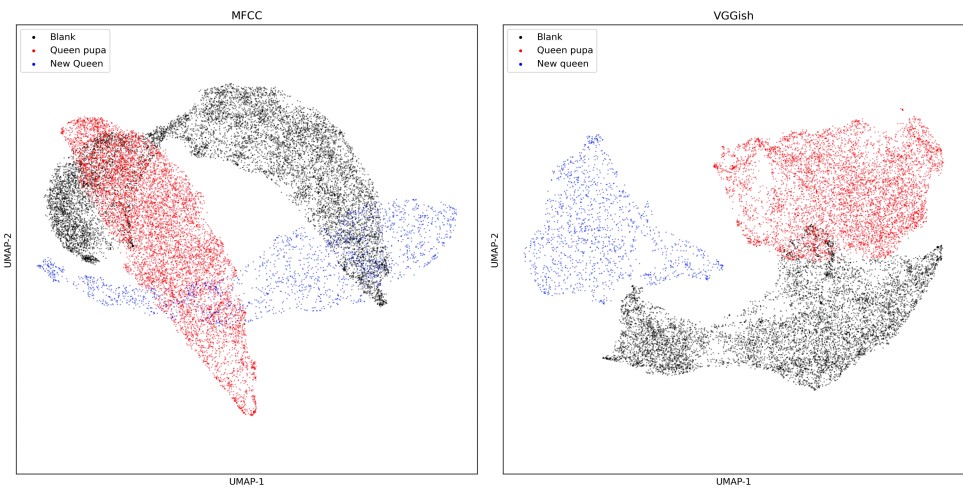

Figure 8  UMAP dimension reduction of sound features from dataset two.

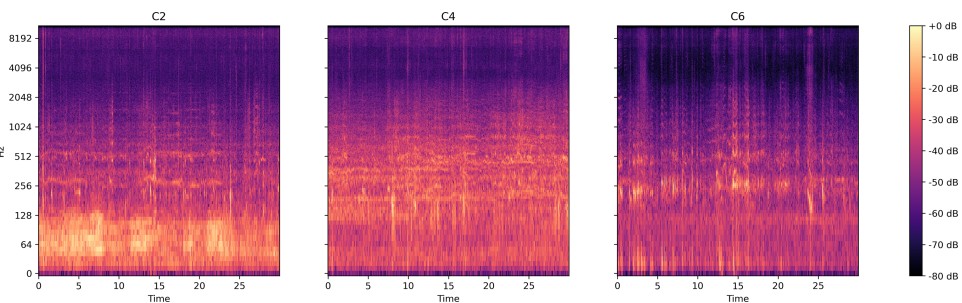

Figure 9  Log spectrum of bee colony sounds for dataset three. Left: Colony size of around 8,500 bees (C2); Middle: Colony size of around 12,500 bees (C4); Right: Colony size of around 19,000 bees (C6).

### *Dimensional reduction of audio feature*

The output of UMAP (Fig. 10) exhibits the VGGish embedding and MFCC of colony sound in dataset three.

### *Model evaluation*

The accuracy of four ML models using different colony sound features on dataset three is shown in Table 3. VGGish embedding has an advantage over MFCC of about 20 percent in all four ML methods, and the difference was statistically significant ($P < 0.05$). Moreover, KNN performed best and achieved an accuracy of 91%.

## The influence of different dimensionality reduction methods

To test the effects of different dimensionality reduction algorithms on the accuracy of the models, we have chosen two-dimensionality reduction algorithms, namely UMAP and t-SNE.

Figure 11 exhibits the results of the dimensionality reduction of dataset one using the t-SNE algorithm, compared with the output of the UMAP algorithm (Fig. 6), UMAP
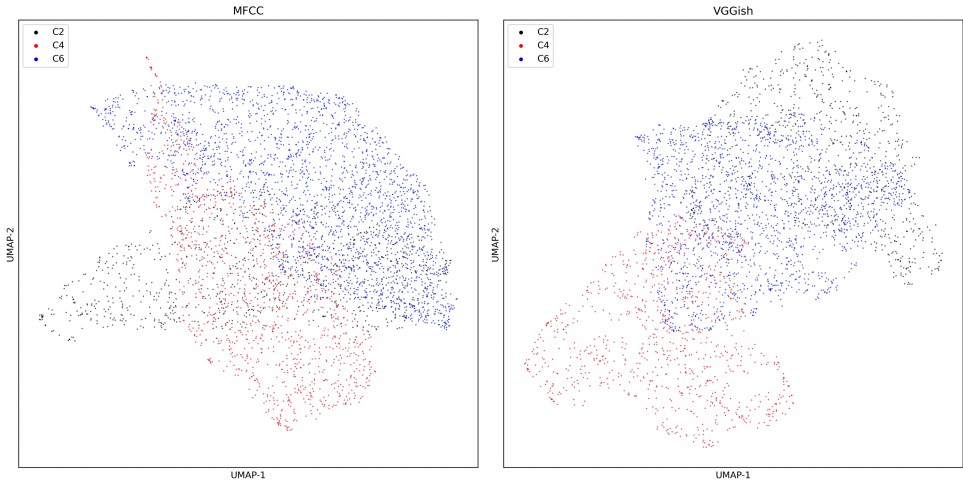

**Figure 10** UMAP dimension reduction of sound features for dataset three.

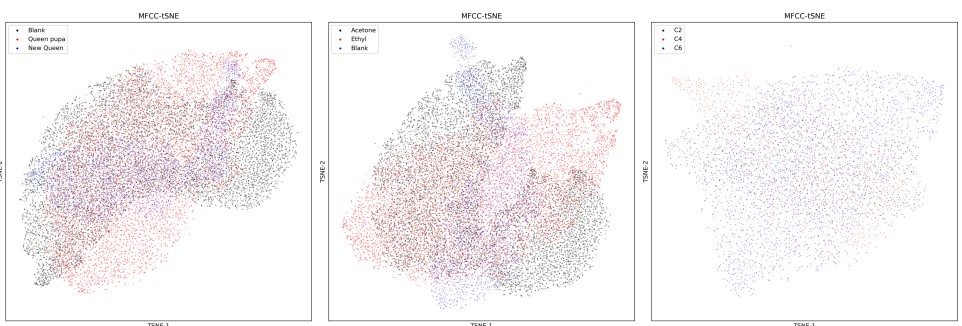

**Figure 11** MFCC features of three datasets after t-SNE dimensionality reduction. Left: MFCC feature using t-SNE dimensionality reduction on dataset two; Middle: MFCC feature using t-SNE dimensionality reduction on dataset one; Right: MFCC feature using t-SNE dimensionality reduction on dataset three.

performs better than t-SNE feature in separating bee colony sounds. Table 4 shows the accuracy of four ML methods trained by two dimension factors obtained by UMAP and t-SNE. The original sound feature used by those dimensional reduction algorithms were the MFCC. The results show that UMAP performs better than t-SNE in almost all datasets and all ML methods.

## DISCUSSION

Hive monitoring based on colony sound has made numerous research achievements in recent years (*Terenzi, Cecchi & Spinsante, 2020*; *Kim, Oh & Heo, 2021*; *Zhao et al., 2021*; *Soares et al., 2022*; *Yu et al., 2022*) and has become increasingly popular with many international companies such as Arnia, Bee Hero, Nectar, and Broodminder (https://www.umt.edu/bee/monitoringconference_2020/).

**Table 5** Comparison of different dimensionality reduction methods.

| Datasets | Dataset 1 | | | | Dataset 2 | | | | Dataset 3 | | | |
|---|---|---|---|---|---|---|---|---|---|---|---|---|
| Algorithm | KNN | DT | RF | SVM | KNN | DT | RF | SVM | KNN | DT | RF | SVM |
| UMAP | 69.09% | 66.28% | 69.17% | 68.29% | 90.48% | 88.45% | 89.95% | 87.25% | 66.04% | 65.78% | 65.13% | 68.05% |
| t-SNE | 51.62% | 54.85% | 55.07% | 56.63% | 62.64% | 65.38% | 66.83% | 66.42% | 52.24% | 55.04% | 57.45% | 60.18% |

In this article, we compared the performance of VGGish embedding and MFCC of bee colony sound in four classification algorithms. Table 3 indicated that all four classification algorithms could generate prediction accuracy percentages that are better than 'chance' based percentages. In all classification methods, the VGGish feature can guarantee more than 80% testing accuracy, among which KNN has the best performance of 94%. The testing accuracy of the MFCC varies greatly between different datasets. In datasets one and three, the MFCC could only achieve an accuracy of about 69%, while in dataset two, it achieved an accuracy of 90%. Results (Tables 3 and 5) show that the difference between the two features in datasets one and three is statistically significant ($P < 0.005$). At the same time, in dataset two, there is no significant difference between the two models ($P > 0.005$).

We confirm that the VGGish embedding applies to bee colony sound classification and performs more stability than the MFCC among different datasets. This may be attributed to the MFCC being highly dependent on data and features which causes weak generalization ability due to insufficient bee colony data and the similarity of bee colony sound. The VGGish network is trained on a more extensive and general Audio set, which means a better generalization ability.

Our results suggest that different compounds do lead to different responses in the bee colony (Fig. 6, Tables 3 and 4), which further confirms the results of previous studies (*Bromenshenk et al., 2009*; *Sharif et al., 2020*; *Zhao et al., 2021*; *Yu et al., 2022*), and verifies the applicability of VGGish embedding in bee colony sound classification. As seen from the log spectrum of bee colony sounds (Fig. 5), the acetone-sucrose solution and acetone ethyl-sucrose solution would agitate the colony compared to the sucrose solution. The low-frequency amplitude was much larger when treated with acetone than when treated with sucrose solution. This may be because acetone stimulates bee colonies more strongly than ethyl acetate at the same concentration, and low concentrations of ethyl acetate were mildly attractive to bees (*Schmidt & Hanna, 2006*).

The MFCC performs better in dataset two (Tables 3 and 4). This may be ascribed to the sound changes fundamentally during bee swarming (*Michelsen et al., 1986*). Thus, it is easier for the standard MFCC to capture the character in colony sounds. Dataset three is small. The total duration of sound in dataset three is less than one hour, and the ML models trained by the VGGish embedding could still achieve an accuracy of around 90%, which may be because the VGGish could better capture the distinctions among the datasets. We have compared two different dimensionality reduction algorithms (Fig. 11, Table 5), and UMAP performs better than the t-SNE in every situation. The secret of UMAP lies in its ability to infer local and global structures while maintaining relative global distances in

low-dimensional space. The result also shows that UMAP performed better in separating different colony sounds.

In summary, the results of this article indicate that the combination of VGGish embedding and the KNN method has achieved the highest accuracy on the testing set of all three datasets (Tables 3–5).

Several ways in which this research can be improved are given below:
(1) Beehive sound samples are few, and only one type of microphone is used for collecting the sound, which causes a lack of data diversity and affects the model's generalizability. A more comprehensive data set must be attained in future work to train the system and improve the model's generalizability.
(2) Expand the application of the model: in this study, we applied VGGish embedding in the classifications of three datasets. Beehive sound can be influenced by many other factors, such as the invasion of natural enemies and parasites. Subsequent studies can check how VGGish embedding performs in these scenarios.

## ACKNOWLEDGEMENTS

The authors would like to thank the Sericultural & Apicultural Research Institute for permission to conduct this study and help during data collection. We also thank Xuewen Zhang, Chuntao Zhou, Chunhui Miao, and Xinqiu Huang, who have generously shared their time and expertise. We appreciate the thorough and thoughtful review by reviewers, their feedback has been extremely helpful in strengthening our argument and improving the clarity of the manuscript.

### Funding

This work was supported by The Hefei Institutes of Physical Science, the Chinese Academy of Science. The funders had no role in study design, data collection and analysis, decision to publish, or preparation of the manuscript.

### Grant Disclosures

The following grant information was disclosed by the authors:
The Hefei Institutes of Physical Science.
The Chinese Academy of Science.

### Competing Interests
The authors declare there are no competing interests.

### Author Contributions

- Nayan Di conceived and designed the experiments, performed the experiments, analyzed the data, prepared figures and/or tables, authored or reviewed drafts of the article, and approved the final draft.

- Muhammad Zahid Sharif analyzed the data, authored or reviewed drafts of the article, and approved the final draft.
- Zongwen Hu performed the experiments, authored or reviewed drafts of the article, and approved the final draft.
- Renjie Xue analyzed the data, authored or reviewed drafts of the article, and approved the final draft.
- Baizhong Yu analyzed the data, authored or reviewed drafts of the article, and approved the final draft.

## Data Availability

The data is available at Zenodo: Nayan Di. (2022, November 8). bee colony sound data. Zenodo. https://doi.org/10.5281/zenodo.7302404.

## Supplemental Information

Supplemental information for this article can be found online at http://dx.doi.org/10.7717/peerj.14696#supplemental-information.

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
