# Peer review of "Applicability of VGGish embedding in bee colony monitoring: comparison with MFCC in colony sound classification"

_PeerJ, doi:10.7717/peerj.14696_

## Round 0.1 · original submission · Major Revisions

All three reviewers pointed out that the manuscript and especially methods and results sections should be substantially improved. Please, consider their comments seriously and submit the new version of the manuscript, which I'd be happy to send to the same reviewers for the second round of review.

·

Basic reporting

Overall, the article was easy to read and professional with only a few minor issues:
Spelling - Coefficients not Cofficients; Numbers nine and below should be spelled out; Frequent and Repetitive use of the adverb Basically that is an unnecessary qualifier; Inclusion of English idioms such as 'In a Nutshell' that can be deleted or replaced by Briefly or Concisely for clarity, and misuse of 'a' or 'an' such as 'A overview rather than the correct 'An overview'. In addition, a beehive is a box, the bees comprise a colony, the colony produces the sounds used to assess the status of the bees (i.e., colony) not of the box (i.e., beehive).

The first two sentences of the manuscript are alarmist and outdated. Clearly there is reason to be concerned about the costs of replacing lost bee colonies, but the beekeeping industry can graft queens and produce bees in quantity. There is no threat of honey bees being lost to the world. The numbers of honey bee colonies worldwide has been growing since 2010. Reduced numbers of bee colonies would affect the diversity of foods readily available. The most likely outcome would be higher costs and more limited selections in grocery stores.

Overall, the literature references with respect to other researchers and users of colony acoustics for periodic or continuous monitoring of honey bee colonies are superficial and fail to properly credit the history and the global extent to which acoustic monitoring of honey bee colonies is already being done. Most of their citations are from the mid-to-late 2000s.

The earliest swarm detection using colony sounds was done by Eddie Woods with his Apidoctor instrument in the U.K. in 1965. In October of 2020, 50 research teams and companies from 14 countries via a live, zoom conference presented their work at the 4th International Bee and Colony Monitoring Conference. Many international companies like Arnia, Bee Hero, Nectar, and Broodminder already have acoustic sensors embedded in their colony monitoring sensor systems. The use of colony sounds to detect colony size, foraging, and queen status is common. The authors cited one of our patents, but no mention was made that we pioneered the detection of a diverse variety of chemicals inside beehives, use both statistical and A.I. (e.g., ANN) analysis of colony sounds, and have a smartphone-based app that citizen scientists around the world since 2019 have been helping calibration of the app to more accurately detect and map colony health parameters.

The authors provided raw data in the form of a dozen wave files from all of the chemical trials, no wave files from the queen trials, and they did not provide the code for VGGish, the various classifiers listed in the Tables, or the MFCC, all of which comprise the tested routines and the focus of this study. The primary purpose of this article was to test whether VGGish embedding and training classifiers and MFCC could be used to improve the extraction of beehive sound features and the identification of colony queen status and the detection and discrimination of two chemicals in nectar. The authors report that MLP performed slightly better. But without the codes, it is unlikely that anyone else could reproduce and confirm the validity of their approach.

Experimental design

The experimental design is inadequately described. For the chemical exposure application, the authors stated that they used four replicates of three groups, presumably one group of colonies that collected syrup spiked with acetone, one group of colonies that collected syrup spiked with ethyl acetate, and a control group.

We have to assume that by the group they meant colonies of bees. They did not provide any information about the health and size of the bee colony populations, the type of hive (e.g., wood, plastic), or whether these colonies were healthy or had any signs of bee pests and diseases. Most importantly, we do not know how many colonies were in each group.

For the queenless versus queen-right colonies, they used colony recordings obtained from an Open Source website. The website states that the downloadable audio files were mostly from novice beekeepers (which tended to be variable in quality) and from a more controlled test based on recordings from only two colonies.

The authors mention obtaining 10-minute recordings, for a total of two hours of data for some of their trials. As with the term group, it's not clear whether the 30 minutes and the two-hour time intervals apply to the total for each individual colony, for a set of group recordings, for four 'replicate' recordings, or for repeated recordings. According to the provided Tables, they obtained four sets of 10-minute recordings for a total of 40 minutes and six sets of 10-minute recordings for a total of 60 minutes.

For the chemical exposure trials, the authors used their own ability to detect the smell of either chemical as evidence that the chemical had dissipated from each hive. The ability of the human nose to perceive the odor of ethyl acetate is about 0.6 mg/ cubic meter and of acetone 0.8 mg/cubic meter. One mg/cubic meter is 0.27 ppm. Since we now know that bees can instantaneously detect odors at parts per quadrillion levels (Alternatives for Landmine Detection, Jacqueline MacDonald et al., 2003, RAND Science and Technology Policy Institute, it is more than likely that there may have been bee-perceivable residual scents from ethyl acetate and from acetone in the hives long after the human investigators could no longer perceive them. Assuming I understand their % numbers for the ehtyl acetate and acetone spikes in syrup (artificial nectar), the syrup collected and brought back into the hives had about 2000 ppm of chemical in the syrup. From our own research, I know that one drop of acetone or toluene elicits a readily perceptible roaring of the colony. It should be noted that 60 minutes of human exposure to 800 ppm acetone or 2000 ppm of ethyl acetate is likely to induce severe illness and other toxic effects.

The methods and algorithms for the VGGish and MFCC that were compared are not described in a manner to enable a reader to understand what was actually tested, especially since the coding was not uploaded for review. The Tables contain accuracy result columns for K-NN, LDA, GNB, and DT, again with no clear definition of these terms nor any upload of the codes and data. The .wav colony sound recordings constitute the measurement data that was collected. The article's focus is on the analytical procedures and output data; which is the assessment data. That data was not provided for review.

The authors appear to assume that they know how other scientists and companies analyze colony sound files. They provide an impression that others use some common form of A.I. They fail to acknowledge the many companies and research groups that currently are engaged in this type of analytical research. There is there any indication that they are aware that others may be using different A.I. approaches from their's, or alternative analysis options such as advanced statistical analyses, data mining, or metadata analytics.

Finally, line 69 mentions the extraction of cellular sound signatures. This phrase is unclear. Bee colonies have wax combs made up of cells; colony monitoring companies use cellular services to transfer data, so in what sense are colony sounds cellular? The phrase implies some aspect of the total sound environment inside a beehive, but it does not define what the authors consider to be a cellular sound signature.

Validity of the findings

Given the experience and background of the authors, I assume that their data analytic's findings are valid. But, it is impossible to answer this question until they provide a better description of their methods, whether they had true colony replication from multiple colonies or just some form of repeated or cumulative recording data from as few as two colonies. And, there is insufficient information to fully review and understand the algorithms and coding that they compared.

Although not clearly stated in the Manuscript Narrative, the Tables indicate replication numbers of 4 or 6, with 4 10-minute recordings for 40 minutes, and six 10-minute recordings for 60 minutes.

As a caution, a common mistake that many honeybee investigators make is an assumption that subsampling a colony or repeated sampling of a colony constitutes proper replication. A colony has only one queen mother. All of the workers and drones have a common mother. The workers may have different drone fathers. The drones do not have fathers. As such, a bee colony is a society made up of sub-families.

Subsampling a bee colony is not replication, and repeated measures of a colony differ from measures of different colonies. As written, it's not clear whether the three groups were replicates, and if so, whether they were adequately replicated.

Additional comments

I sincerely hope that the authors revise the manuscript, especially by better defining their methods, providing the number of colonies (i.e., replication), and uploading the code for the extraction and analytical procedures.

They did confine their conclusions to their own research and results, but they did not show whether they found a method that was as good or better than the approaches being used by other research groups or companies.

The two chemicals that they used may or may not be osmophores. We can not assume that just because acetone may be irritatingly sweet to some people, and that ethyl acetate has a sweet or pineapple flavor that is how these compounds 'smell' or taste to bees. Both chemicals are common industrial solvents. These two chemicals are just two of over 200 semi-volatile and volatile organic chemicals, in addition to several hundred pesticides, that may be found in beehive atmospheres (2003 monograph review of Honey Bees: Estimating the Environmental Impact of Chemicals, Jame DeVillers, and Minh-Ha Pham-Delegue).

Given the volatility of acetone, one can not assume that acetone in nectar is being detected by the bees in the colony due to ingestion. When the syrup from the honey crop of a forager is regurgitated in the hive, the acetone should volatilize directly into the hive atmosphere. Ethyl acetate is less volatile, but since it was easily smelled by the researchers, there must have been ethyl acetate in the hive atmosphere.

My point is that the study did not adequately address the question of whether introducing contaminants into the hive via syrup was any different from injecting gases directly into the hive with respect to inducing a colony sound response. The implication seems to be that the authors assumed that the bees would be acoustically responding to the presence of these two chemicals in transported, ingested, or stored nectar. Frankly, the dose of chemicals in nectar collected by bees was extreme.

What this investigation did accomplish was a confirmation that two chemicals and the queen status of a colony could be reliably detected based on analysis of colony sounds. Several others have reported the same, adding up to nine other colony status assessments in addition to the presence or absence of the queen and many other chemicals.

Regardless, there is value in confirmatory studies, and any improvement, even incremental, in the ability to accurately classify colony status and chemical recognition has value. But that value to the science is dependent upon making their analytical decision routines readily available for use.

Reviewer 2 ·

Basic reporting

1) The text of the article has numerous grammatical errors that must be corrected.

2) The related work mentioned in the article is dated (e.g., 2008 or 2013). There has been a lot of progress in audio-based beehive monitoring since then, which the authors do not seem to be aware of. These publications contain additional important references to audio hive monitoring.

a) S. Cecchi, S. Spinsante, A. Terenzi and S. Orcioni, “A Smart Sensor-Based Measurement System for Advanced Bee Hive Monitoring”, Sensors 2020, 20(9), 2726; https://doi.org/10.3390/s20092726.

b) M. Bencsik, M. Newton, I. Michael, M. Reyes, M. Pioz, D. Crauser, N. Delso, Y. Le Conte, "The prediction of swarming in honeybee colonies using vibrational spectra,” Scientific Reports, vol. 10, 2020.

c) V. Kulyukin, S. Mukherjee and P. Amlathe, “Toward Audio Beehive Monitoring: Deep Learning vs. Standard Machine Learning in Classifying Beehive Audio Samples,” Applied Sciences, 2018, 8(9), 1573;
https://doi.org/10.3390/app8091573.

d) C. Gupta. "Feature selection and analysis for standard machine learning classification of audio beehive samples." M.S. Thesis. Department of Computer Science, Utah State University, Logan UT, USA, 2019.

e) A. Bhouraskar. "Automation of feature selection and generation of optimal feature subsets for beehive audio sample classification." M.S. Thesis. Department of Computer Science, Utah State University,
Logan UT, USA, 2020.

3) Several audio files are included. Figures are relevant. No picture of the hardware setup or the actual deployment of the system.

4) They hypotheses are not stated. Lines 241-242: The pollutants in nectar can be perceived by the colony and show up in changes in the beehive sound.

This is not a valid conclusion w/o ANOVA. Another concern is insufficient data: the identification of
compounds included only 40 minutes of audio data and the identification of queen presence was done on
60 minutes. How many of those were recorded by the authors?

Experimental design

1) Line 1: In recent years, the population of honeybee (Apis mellifera L.) has been declining in recent years (Spleen et al., 2013).

This sentence is grammatically incorrect. Two more points about the content of this sentence. First, the 2013 reference is a dated reference. More recent investigations can be cited. Second, the authors should take a look at
https://www.acsh.org/news/2018/04/17/bee-apocalypse-was-never-real-heres-why-12851.
The bee apocalypse is frequently hyped to get funding/attention.

2) Line 7: Frequent interference is a stressor that brings anxiety to the swarm.

This sentence is not conceptually correct: the swarm is a technical term in apiary science where one
queen (typically, the older one) leaves the colony with approximately half of the foragers; the other half stays behind with the new queen.

3) The two main objectives of the study are stated in lines 67 -- 68:

"The main purpose of the study was twofold: to test whether VGGish could effectively
extract cellular sound signatures, and to verify whether the colony could sense the presence
of contaminants in nectar."

Effectiveness must be statistically significant which can be found with ANOVA. Giving
only percentages is not meaningful with statistical significance.

4) Lines 77 -- 79: The experiment was carried out in an open space in the Sericulture and Apiculture Research Institute of Yunnan Academy of Agricultural Sciences, from November 2020 to January 2021.
Three groups were set up: two experimental groups and one control group.

This is an inadequate description of the experimental setup. What is a group? How many hives did it include? What queen lines were used? How old were the queens? Where was the apiary located? Was it urban or rural? These questions are never answered.

5) Lines 82: In the first few days of the experiment, we marked the foraging bees with a yellow marker.

Were the foragers marked in each group?

6) Line 86: Marked foraging bees were trained to visit at the feeder.

How were they trained?

7) Line 89: The three groups alternate randomly.

What does it mean? Why does this sentence use the past tense?

8) Lines 93 - 94: We collected some of the data from our own beehives while other was obtained from
different sources including global researcher.

Lines 104--105: A microphone (Neutrik, Switzerland) was placed on top of the beehive, and a digital
sound card (UM2, BEHRINGER) was used.

Where exactly was the microphone placed? On top of the hive? Did you collect ambient sound then?
Was it placed inside the hive or outside of the hive? A picture of the setup is a must. What computer
was the mike connected to? These questions are not answered in the article.

9) Lines 111-12: A microphone (Neutrik, Switzerland) was placed on top of the beehive, and a digital
sound card (UM2, BEHRINGER) was used.

So, each wav was passed through the Google CNN to extract the features. A picture/diagraph would be
nice here showing the architecture of the convolutional network.

10) Lines 114-115: The MFCC feature extraction procedures refer to our previous research, basically
include windowing the sound signal...

The MFCC were invented in the 1980's. It's a very well known technique and a reference must
be included in the sentence.

11) The numbers reported in Section 3.2 don't make much sense in the absence of the control groups sizes and tests of statistical significance. Without these statistics the article cannot be considered a valid contribution.

12) Lines 238-239: 1) VGGish embedding can be used to effectively identify beehive status and achieve good results under various classification algorithms.

This is not an accurate sentence, because the authors never defined "status." What is "status"? What is "effectiveness"?

Validity of the findings

1) In and of itself, the article is not novel. MFCCs and convolutional networks have been used in audio beehive monitoring. The authors did not give sufficient rationale why their replication is a contribution.

2) Several audio files were attached. It is unclear if these files constitute the entire dataset.

3) Lines 241-242: The pollutants in nectar can be perceived by the colony and show up in changes in the beehive sound.

This is not a valid conclusion without careful statistical analysis. Another concern is insufficient data: the identification of compounds included only 40 minutes of audio data and the identification of queen presence was done on 60 minutes. How many of those were recorded by the authors?

Additional comments

No comment.

Reviewer 3 ·

Basic reporting

Although in general the structure of the paper is correct, it is necessary to restructure and expand the content of some sections so that they comply with what they are intended to contain. The descriptions of the methods are poor, as well as the results and conclusions sections.

Experimental design

Methods
When the algorithms DT, MLP, KNN, LDA, and GNB are mentioned for the first time, the meaning of the acronyms must be mentioned, for example: Decision Tree (DT), just as it is done in line 133.
The target of the classification must be indicated. Later, in Results, it is mentioned that the tests are performed to identify the presence of the queen in the hive and the identification of simulated nectar contaminants.

Introduction
2: Text "in recent years" duplicated in the phrase.
30-34: Although when mentioning that machine learning techniques are used, the fact of training the model can be taken for granted, it is important to indicate in the process the importance of the training phase of the classification model and therefore the data set used for that purpose
50-66: A table can be used to show in a clearer way the architecture of the Neural Network used.
At the end of the Introduction section it is recommended to include a paragraph mentioning what is covered in the remaining sections of the document.


Material and Methods
The name of the section should be "Materials and Methods".
77-78: Complete the information of the place where the experiment was carried out, indicating the country.
77-81: More context should be provided on the conditions of the experimental and control groups, only the quantities used are mentioned, but the explanation could be expanded, indicating that it is a substance used as artificial nectar.
92-96: Being a job where the importance lies in the data used, the percentage of data collected on its own and those obtained from other sources must be specified.
97-101: The data set used is a very important part of the work, so it must be described in greater detail.
102-107: Justify the choice of hardware and software used for the process of obtaining the sound
109-113: Use a diagram to better describe the process mentioned. Also justify the choice of values ​​for the mentioned parameters.
114-119: Use a diagram to better describe the process mentioned
132-140: Although they are widely known algorithms in the area, to have a correct structure they must be described indicating their operation and main characteristics.
In general, this section is poorly described, it should be complemented to better support the proposed work


Results
Some of the results obtained are only mentioned again, but their impact or relevance to the proposed work is not discussed. A deeper analysis of the results obtained is necessary.


Conclusion
MFCC features are not mentioned in the conclusion, so there is nothing to conclude about them?
The conclusions are poor and very general, they do not highlight the contribution of the proposed work. It is necessary to rewrite this section, expanding it and carrying out a better analysis of the proposal made and the results obtained.

Validity of the findings

Methods
When the algorithms DT, MLP, KNN, LDA, and GNB are mentioned for the first time, the meaning of the acronyms must be mentioned, for example: Decision Tree (DT), just as it is done in line 133.
The target of the classification must be indicated. Later, in Results, it is mentioned that the tests are performed to identify the presence of the queen in the hive and the identification of simulated nectar contaminants.

Introduction
2: Text "in recent years" duplicated in the phrase.
30-34: Although when mentioning that machine learning techniques are used, the fact of training the model can be taken for granted, it is important to indicate in the process the importance of the training phase of the classification model and therefore the data set used for that purpose
50-66: A table can be used to show in a clearer way the architecture of the Neural Network used.
At the end of the Introduction section it is recommended to include a paragraph mentioning what is covered in the remaining sections of the document.


Material and Methods
The name of the section should be "Materials and Methods".
77-78: Complete the information of the place where the experiment was carried out, indicating the country.
77-81: More context should be provided on the conditions of the experimental and control groups, only the quantities used are mentioned, but the explanation could be expanded, indicating that it is a substance used as artificial nectar.
92-96: Being a job where the importance lies in the data used, the percentage of data collected on its own and those obtained from other sources must be specified.
97-101: The data set used is a very important part of the work, so it must be described in greater detail.
102-107: Justify the choice of hardware and software used for the process of obtaining the sound
109-113: Use a diagram to better describe the process mentioned. Also justify the choice of values ​​for the mentioned parameters.
114-119: Use a diagram to better describe the process mentioned
132-140: Although they are widely known algorithms in the area, to have a correct structure they must be described indicating their operation and main characteristics.
In general, this section is poorly described, it should be complemented to better support the proposed work


Results
Some of the results obtained are only mentioned again, but their impact or relevance to the proposed work is not discussed. A deeper analysis of the results obtained is necessary.


Conclusion
MFCC features are not mentioned in the conclusion, so there is nothing to conclude about them?
The conclusions are poor and very general, they do not highlight the contribution of the proposed work. It is necessary to rewrite this section, expanding it and carrying out a better analysis of the proposal made and the results obtained.

---

## Round 0.2 · Major Revisions

For preparing your revised version, please, check the comments of both reviewers.

·

Basic reporting

The revised manuscript is massively improved. I still picked up a lot of errors using MS Word Editor and Grammarly. I will attach my markup of the manuscript. Literature references are considerably better, but they still don't cover much of anything from more than a few years past. More concerning is that some of the statements in the manuscript don't match what's in the citations. It's not sufficient just to read a title or a summary of a citation. I suggest that the editor ask the authors to verify the actual information in cited articles and patents.

The professional appearance and structure are much improved, and the raw data is shared. The manuscript now is self-contained and stands on its own.

Experimental design

This time, the authors used their own colonies, rather than data sets from YouTube of unknown provenance. They still reference YouTube, which may be an oversight.

The research questions have been more clearly stated, it's now clear what they did, and they provide a more honest assessment of the significance of the research.

It appears that there is no sufficient information to replicate, but there are some very confusing statements. In the previous version of this manuscript, they tested the two routines (models) using data from large colonies of Apis mellifera, the European honey bee that has been introduced to countries around the world. Virtually all previous acoustic investigations have focused on A. mellifera. However, the authors say that they conducted their investigations using the smaller A. cerana, the Asian honey bee. Yet the numbers of bees per frame and the average weight per bee suggest that they were actually using Apis mellifera. If they tested using A. cerana, then this is one of or is the only work using that species of honey bees. If they investigated A. mellifera, they need to correct the name used in the manuscript.

They talk about a reduction of the acquired signals from a 44 kHz to 16 kHz with no explanation for the need to do this. They also imply a 30kHz to 20kHz recorded signal, but it's more likely that they meant to say that their microphones and soundcard recorded a 30 Hz to a 20 kHz range. Again, I ask that they review and correct as appropriate.

All in all, they are still using very high concentrations of chemicals in the treatments. Bees can perceive scents for most chemicals in the parts per trillion range. If you over-dose them, one can hear the change in colony sound by ear alone.

In addition, they are still basing their findings on very small replication - three colonies. They then use the majority of the recordings for training, and only reserve about 1/3 for testing. That approach is weighted for success when the replication size is only three colonies.

I leave it to my fellow reviewer to address the final statistics. I already question the use of a Student T test for acoustic recordings that have a very diverse set of characteristics.

Validity of the findings

Replication is small. The underlying data has been provided. Conclusions are balanced, admitting that the question was whether VGGish or MFCC alone or together improved discrimination ability and accuracy.

Overall, this work represents possible incremental improvements in the identification of factors such as exposure to two chemicals or queen status by using these routines when analyzing the data.

For the many groups around the world struggling to identify colony health and to detect chemical exposure incidents, any improvement is important.

Furthermore, the other research groups could not have replicated the approach as described in the original manuscript, but they should be able to do so now, although they may need help from the authors in terms of implementation.

Additional comments

I recommend publishing after corrections of errors, better proofing of the English, and re-checking of the work and findings of the cited literature and patents.

I think they were testing using A. mellifera, not A. cerana. These are very different bees, and most of the existing research has been done using A. mellifera, not A. cerana.

The frequency ranges recorded need to be made clear, and the rationale for using a Coefficient normally applied to human speech, which is very different from bee colony sounds, isn't clear.

Reviewer 2 ·

Basic reporting

It's been improved. It reads better than the previous version.

Experimental design

The authors state that "The hardware and software systems for obtaining bee colony sound are as follows: A microphone inside the beehive (PCK200, TAKSTAR) was placed about 15cm from the bottom. The microphone has a frequency range of 30-20 kHz and a sensitivity of -35 dB."

Did the bees propolise the microphone? Did the authors have to remove the propolis?

Validity of the findings

The amount of data is very rather small. Therefore, it's hard to tell whether the findings are valid or not. The article reads like a pilot study.

Additional comments

no comment

---

## Round 0.3 · Minor Revisions

The Section Editor noted:

> The title is not comprehensible for a reader not working with bees. I suggest improving it

Please edit the title to make it clear to a wider audience.

---

## Round 0.4 · accepted · Accept

The author provided appropriate changes and therefore I accept this version of the manuscript.